# Involvement of PI3K/HIF-1α/c-MYC/iNOS Pathway in the Anticancer Effect of *Suaeda vermiculata* in Rats

**DOI:** 10.3390/ph16101470

**Published:** 2023-10-16

**Authors:** Hamdoon A. Mohammed, Mohamed G. Ewees, Nesreen I. Mahmoud, Hussein M. Ali, Elham Amin, Mohamed S. Abdel-Bakky

**Affiliations:** 1Department of Medicinal Chemistry and Pharmacognosy, College of Pharmacy, Qassim University, Buraydah 51452, Saudi Arabia; ham.mohammed@qu.edu.sa; 2Department of Pharmacognosy and Medicinal Plants, Faculty of Pharmacy (Boys), Al-Azhar University, Cairo 11884, Egypt; 3Department of Pharmacology and Toxicology, Faculty of Pharmacy, Nahda University, Beni-Suef 11787, Egypt; mohammed.gamal@nub.edu.eg (M.G.E.); nesreen.mahmoud@nub.edu.eg (N.I.M.); 4Department of Pharmacology and Toxicology, College of Pharmacy, Qassim University, Buraydah 51452, Saudi Arabia; hu.ali@qu.edu.sa; 5Department of Biochemistry, Faculty of Medicine, Al-Azhar University, Assiut 71524, Egypt; 6Department of Pharmacognosy, Faculty of Pharmacy, Beni-Suef University, Beni-Suef 62514, Egypt; 7Department of Pharmacology and Toxicology, Faculty of Pharmacy, Al-Azhar University, Cairo 11884, Egypt

**Keywords:** *Suaeda vermiculata*, antioxidant, hepatocellular carcinoma (HCC), PI3K, HIF-1α, c-MYC, iNOS, N-diethylnitrosamine (NDEA)

## Abstract

*Suaeda vermiculata* Forssk. ex JF Gmel. (SV), a traditional known plant, has shown in vitro cytotoxic activity against HepG2 and HepG-2/ADR (doxorubicin-resistant cells) liver cell carcinoma cell lines, as well as hepatoprotection against paracetamol and carbon tetrachloride (CCl4)-induced liver injury. The current study evaluated the protective effect of SV, administered against N-diethylnitrosamine (NDEA)-induced HCC in rats. The possible modulatory effect of SV on the PI3K/HIF-1α/c-MYC/iNOS pathway was investigated. Sixty male adult albino rats (200 ± 10 g) were equally classified into five groups. Group I served as a control; Group 2 (SV control group) received SV (p.o., 200 mg/kg body weight); Group 3 (NDEA-administered rats) received freshly prepared NDEA solution (100 mg/L); and Groups 4 and 5 received simultaneously, for 16 weeks, NDEA + SV extract (100 and 200 mg/kg, orally). NDEA-treated rats displayed significant increases in serum levels of AFP, CEA, PI3K, malondialdehyde (MDA), epidermal growth factor receptor (EGFR), and vascular endothelial growth factor (VEGFR), with increased liver tissue protein expression of fibrinogen concomitant and significantly decreased concentrations of antioxidant parameters (catalase (CAT), superoxide dismutase (SOD), and reduced glutathione (GSH)) in comparison to normal rats. On the flip side, AFP, CEA, PI3K, MDA, EGFR, and VEGFR serum levels were significantly reduced in rats that received NDEA with SV, both at low (SV LD) and high (SV HD) doses, accompanied by significant improvements in antioxidant parameters compared to the NDEA-treated group. Conclusions: SV possesses a significant hepatoprotective effect against NDEA-induced HCC via inhibiting the PI3K/HIF-1α/c-MYC/iNOS pathway, suggesting that SV could be a promising hepatocellular carcinoma treatment.

## 1. Introduction

The liver is an important soft tissue responsible for several functions in the body, including the metabolism of food as well as the detoxification of drugs and toxins [1]. It plays a pivotal role in the immune, coagulation, and digestion systems [1]. Liver cells, including hepatocytes, are considered sensitive cells to toxins and oxidizing agents, which are considered inducers for several hepatic diseases such as liver cancer, fibrosis, cirrhosis, and other dysfunctions [2,3]. Due to the sensitivity of liver cells to toxins and oxidizing agents, under certain circumstances, liver supports become essential protectors that compensate for the reduction in or depletion of the indigenous antioxidant factors of the body, such as reduced glutathione (GSH) as well as catalase (CAT), superoxide dismutase (SOD), and glutathione reductase (GR) activities [4]. Liver support also reduces and prevents the progress of the inflammatory process that may increase liver cell proliferation and cancer development [5]. Natural products are the most popular medications used for liver support. This is due to their powerful antioxidant effect, which is attributed to their content of natural antioxidants such as phenolic acids and flavonoids. Therefore, silymarin, the most useful drug in the treatment of liver dysfunction, is a natural plant complex of flavonolignans obtained from milk thistle, *Silybum marianum* [6]. In addition, several plants or plant parts are commonly used in folk medicine for liver health [7,8,9,10]. Furthermore, several species of plants have been investigated for their potential hepatoprotective effect, along with their mechanism of action and the phytochemicals involved in that effect [11,12,13]. The results of those investigations have enriched the literature with potential hepatoprotective plant-based materials.

Herein, we are continuing our previous investigations on the liver-protecting activity of *Suaeda vermiculata* Forssk. ex JF Gmel. (SV), (Amaranthaceae), owing to its use in traditional medicine for similar effects [14,15]. The plant is widely grown in the marsh areas of central Saudi Arabia and is considered a halophyte due to its ability to survive in the relatively salted areas. In previous research, SV fractions obtained from the acholic extract of the plant exerted substantial in vitro cytotoxic activity for HepG2 and HepG-2/ADR. This activity was credited to the flavonoids and phenolic acid contents of this species, which exist as a mixture of compounds such as coumaric acid, chlorogenic acid, spiraeoside, hyperoside, kaempferol glycosides, luteolin glycoside, vitexin, and quercetin aglycone and glycosides [16]. The plant extract has also demonstrated a hepatoprotective effect on paracetamol and CCl4-induced oxidative damage in mice and rat models, respectively [17,18]. In addition, two pure flavonoids (isorhamnetin-3-O-rutinoside and quercetin) and one chlorophyll-based (Pheophytin-A) natural product have been isolated from SV and exerted potential liver protection, demonstrated by lowering liver toxicity markers and restoring the antioxidant enzyme levels disturbed by paracetamol in mice [17].

As a continuation of the previous work, the current research focuses on the in vivo hepatoprotection and anti-hepatocellular carcinoma effects of SV. It was designed to study the effect of SV against N-diethylnitrosamine (NDEA)-induced oxidative stress and liver cancer in rats. In addition, the possible modulatory effect of SV on the Phosphoinositide 3-kinases (PI3K)/Hypoxia-inducible factor 1-alpha (HIF-1α)/c-MYC/Nitric oxide synthase (iNOS) pathway involved in NDEA-induced HCC fibrinogen, together with the possible interrelation with the coagulation system activation marker, was also demonstrated.

## 2. Results

### 2.1. Effect of SV, in the Presence or Absence of NDEA, on AFP, CEA, and PI3K Serum Levels

The results indicated that NDEA-treated rats displayed significant elevations (at *p* < 0.001) in AFP (291.8%), CEA (257.7%), and PI3K (343.8%) serum levels compared to normal non-treated rats. On the other hand, the serum levels of AFP (66.4 and 62.6%), CEA (69.8 and 65.6%), and PI3K (55.2 and 42.3%) displayed a significant reduction (at *p* < 0.001) in NDEA-treated rats and co-treated with SV at high (SV HD) and low (SV HD) doses (100 and 200 mg/kg, respectively) when compared to rats receiving only NDEA. Although the SV HD and SV HD reduced the levels of AFP, CEA, and PI3K induced by the NDEA, the levels did not normalize when compared to the normal group. This was demonstrated by the significant differences (at *p* < 0.001) in SV HD- and SV HD-treated groups when compared to normal rats. Furthermore, serum levels of AFP, CEA, or PI3K in SV-treated animals displayed no significant changes compared to normal rats (Figure 1).

### 2.2. Effect of SV, in the Presence or Absence of NDEA, on the Liver Tissue Levels of MDA, CAT, SOD, and GSH

The anti-oxidants (GSH, SOD, and CAT) and the end products of polyunsaturated fatty acids peroxidation, MDA, were analyzed in liver homogenates using the ELISA technique. Results demonstrated in Figure 2 indicated that NDEA treatment induced a significant (at *p* < 0.05) level of MDA to 41%, which was concomitant with reduced (at *p* < 0.05) anti-oxidant markers in the present study, i.e., CAT (28.8%), SOD (31.2%), and GSH (30.15%), as compared with control non-treated rats. Conversely, the NDEA-co-treated rats with SV HD and SV LD displayed a significant reduction (at *p* < 0.05) of MDA to 50.8% and a significant elevation (at *p* < 0.05) in the anti-oxidant markers CAT (244 and 308%, respectively), SOD (245 and 283%, respectively), and GSH (261 and 294%, respectively) compared to the rats that received only NDEA. In addition, non-significant differences were observed in MDA and anti-oxidant markers between SV-only-treated rats, except for a significant reduction in GSH, compared to the control non-treated group and between NDEA + SV HD and NDEA + SV HD-treated groups (Figure 2A–D).

### 2.3. Effect of SV on EGFR and VEGFR Serum Level and Gene Expression of c-MYC and HIF-1α in the Presence or Absence of NDEA

Levels of EGFR and VEGFR in the serum were analyzed using the ELISA technique. Serum EGFR (642%) and VEGFR (533%) were increased significantly (at *p* < 0.05) in the NDEA-treated group in comparison with the normal animals. Moreover, the NDEA + SV HD and NDEA + SV HD groups displayed a marked decline (at *p* < 0.05) in the serum levels of EGFR (55.7 and 42.4%, respectively) and VEGFR (55.3 and 42.5%, respectively) compared to NDEA-alone-treated rats. No changes were found in the serum levels of EGFR and VEGFR in the SV-treated rats in comparison to normal rats. In addition, no marked changes were detected in EGFR and VEGFR serum levels between the high and low doses of SV + NDEA-treated groups (Figure 3A,B).

The gene expressions of HIF-1α and c-MYC were assessed using the RT-PCR technique. The results in Figure 3C,D reveal an elevation in HIF-1α expressions (387.5%) and c-MYC (292.8%), respectively, in the NDEA-administered group in comparison to the normal group. This effect was diminished when NDEA was administered with SV (200 mg/kg or 100 mg/kg), as indicated by the significantly lower expressions of HIF-1α (40.3 and 30.6%, respectively) and c-MYC (56.09 and 46.3%, respectively) in the presence of NDEA in comparison to the NDEA-treated rats. Moreover, expressions of HIF-1α and c-MYC were not significantly changed after treatment with SV alone compared to control, non-treated rats. In addition, no marked changes were seen in HIF-1α or c-MYC expressions between the NDEA + SV HD and NDEA + SV HD-treated groups (Figure 3C,D).

### 2.4. Effect of NDEA with or without SV on Fibrinogen Protein Expression

Protein expression of fibrinogen in liver tissues of normal and SV-administered rats showed basal expression in the endothelial cells of the central vein and in the peri-central hepatocytes. On the other hand, NDEA-treated rats showed intensive expression of fibrinogen in the hepatic nodules, large nodules of acinar HCC, and peri-central hepatocytes. Rats treated with both NDEA + SV HD and NDEA + SV HD displayed marked reductions in the basal levels of fibrinogen expression, which were observed in the peri-central zone and peri-central hepatocytes compared to the rats receiving NDEA. There were no marked differences in fibrinogen expression between SV treatments at high and low doses (Figure 4A–C).

### 2.5. Effect of SV on iNOS Proteins Expression in Liver Tissues with or without NDEA

The results in Figure 5 indicated negative expression of iNOS either in the control, untreated, or SV groups. However, the expression of iNOS was detected in the NDEA-treated group in hepatic nodules and large nodules of acinar and peri-central hepatocytes. On the other hand, a noticeable reduction in iNOS was seen in the co-administration of NDEA + SV HD and NDEA + SV HD (Figure 5A,B).

## 3. Discussion

The current study continues our prior investigation of the biological efficacy of *S. vermiculata* Forssk. ex JF Gmel., an edible halophyte widely distributed throughout the central region of Saudi Arabia. The plant was previously noted for its hepatoprotective and anti-HCC effects as part of its traditional use in the treatment of various liver illnesses [16,18]. These bioactivities were suggested to be attributed to the plant’s abundant flavonoids and phenolic contents. The metabolic content of SV was previously exhaustively investigated by our group, where the phenolic and flavonoid contents in this species were quantitatively estimated as 200.6 ± 2.14 GAE and 16.70 ± 1.56 QE, respectively [18]. Furthermore, the phytoconstituents of SV were chromatographically explored using variable chromatography techniques. HPLC-QTOF-MS/MS analysis revealed diversified metabolites from different chemical classes, such as flavonoids (e.g., isorhamnetin, kaempferol, luteolin, quercetin, and their glycosides), phenolic acids (e.g., coumaric acid and chlorogenic acid), fatty acids (hexadecanoic acid, linoleic acid, octadecanoic acid, palmitic acid, and elaidic acid), etc. [16,17,18]. Moreover, our previous study recorded the isolation of three chemical components: isorhamnetin-3-O-rutinoside, pheophytin-A, and quercetin from SV and concluded their significant reduction in oxidative stress and inflammatory indicators as LP, NO, IL-6, and TNF-α. They also restored the SOD and CAT enzyme levels in the liver-injured rats’ models [17].

HCC accounts for approximately 80–90% of all cancer liver types, with a low survival rate as most cases of HCC are not diagnosed in the early stages and, therefore, its treatment becomes challenging [19]. HCC ranks sixth in the world in terms of prevalence and fourth in types of cancer leading to death [20]. It can be said that exposure to aflatoxin B1, alcohol abuse, obesity, diabetes, and hepatitis C and B viruses are among the important causes of hepatic cancer. These cause hepatitis, fibrosis, and cirrhosis of the liver, which results in increased oxidative stress followed by apoptosis of the liver and consequently HCC development [21].

In the current work, we found that injecting the rats with NDEA resulted in HCC induction, increased liver oxidative stress, and reduced endogenous antioxidants. The effect was reduced by using high and low doses of SV. The results revealed a decreased antioxidant defense system in NDEA treatment, as indicated by decreased glutathione (GSH), catalase (CAT), and superoxide dismutase (SOD) levels. In contrast, NDEA increased the oxidative stress marker, malondialdehyde (MDA). Previous reports have shown the accumulation of reactive oxygen species (ROS) inside cancerous cells with excessive levels of free radicals and thus a defect in antioxidant production [22,23]. ROS play an important role in different types of cancer development [24]. Furthermore, NDEA increased oxidative stress through hepatic metabolism via the cytochrome p450 enzyme [25]. Ethyldiazonium ion alkylates nuclear DNA bases through NDEA to form adducts such as O4 and O6-ethyldeoxythymidine and O6-ethyldeoxyguanosine [26]. Moreover, oxidative stress is increased in diabetes and obesity and has an important role in HCC development since it causes gene instability and damage [27]. In addition, a reduction in the biosynthesis of GSH has been found during HCC injury [28]. This reduction could be attributed to its massive consumption during the free radical scavenging process induced by the liver metabolism of NDEA. The increased free radical production resulting from NDEA metabolism may be the cause of decreased antioxidant enzyme activities. In the current work, increased free radical and MDA production was associated with a reduction in antioxidant enzymes, suggesting that radical formation reduces antioxidant enzyme activities [29]. Increased GSH, SOD, and CAT antioxidant enzyme levels and decreased MDA levels by SV treatment disturbed by NDEA-treated rats suggested its protective effect, which was mediated at least in part through the induction of antioxidant mechanisms and reduction in free radical formation.

In the current work, we found increased levels of VEGFR and EGFR in NDEA-induced HCC in rats, which decreased significantly by using both SV doses (200 and 100 mg/kg) compared to NDEA-treated rats. EGFR is a receptor that binds to the hepatocytes as a mitogenic agent, transforming growth factor alpha (TGF-a) [30]. Previous research has reported the involvement of the TGF-a/EGFR pathway in the development of HCC, in which TGF-a messenger RNA (mRNA) expression is significantly increased [31]. Furthermore, using an EGFR-specific inhibitor decreased the onset of HCC in other studies [32]. Additionally, VEGF mediates angiogenesis during tumor growth through mitogenesis induction and vascular permeability [33]. VEGF has two main receptors, VEGF receptor-1 and 2 (VEGFR-1 and VEGFR-2) [34]. It has been found that the protein expression as well as upregulated mRNA levels of VEGFR-2 were increased in HCC [35]. Efforts have been made to use inhibitors for VEGFR, such as Sorafenib, to control tumor vascular growth [36]. For that, the level of VEGFR expression is directly associated with the efficiency of anti-angiogenic treatments [35]. Notably, SV significantly inhibited EGFR and VEGFR induced by NDEA treatment, suggesting the possibility that both receptors could play a role in the SV hepatoprotective effect.

Phosphatidylinositol 3-kinase (PI3K)/protein serine threonine kinase (Akt)/mammalian rapamycin (mTOR) signaling pathway mutations and distressed expression have been observed in HCC [37]. Our data highlight the role of PI3K, HIF-α, iNOS, and c-MYC in HCC development and the possible modulatory effect of SV on PI3K, HIF-α, iNOS, and c-MYC in NDEA-induced HCC. The signaling pathway of PI3K/Akt/mTOR contributes to cell proliferation, apoptosis, cell cycle regulation, angiogenesis, and metabolism. In different cancer types, the activity of this pathway has been observed [38]. Moreover, Glycogen Synthase Kinase-3 (GSK-3) mediates a sequence of substrates such as c-MYC and HIF-1α to induce cell growth and tumor development [39]. Nitric oxide (NO) produces inducible nitric oxide synthase (iNOS), which is considered an important signal in tumors and has a controversial mechanism during cancer development to either inhibit or promote the growth of the tumor [40]. Additionally, iNOS is correlated with the low survivability of breast cancer patients by enhancing the aggressiveness of the tumor [41]. Notably, the signaling of NO triggers the activity of PI3K/Akt-dependent c-MYC and finally induces tumor cell proliferation [38]. Adaptation of the malignant tumors to hypoxic stress is rolled out by HIF-1 during tumor vascularization and invasion. In addition, HIF-1 contributes to the induction of iNOS under hypoxic conditions [42]. This study, for the first time, reports the hepatoprotective effect of SV against HCC induced by NDEA in rats, suggesting that this effect is mediated at least in part through the PI3K/HIF-α/c-MYC/iNOS pathway.

Deposition and clearance of fibrin are continuously occurring during cancer development [43]. An increased level of fibrinogen has been associated with the progression of different malignancies [44]. Fibrinogen has been elevated after cancer development and, therefore, may serve as an effective and simple marker for the bad prognosis and outcomes of HCC [45]. To the best of our knowledge, no data showing a link between fibrinogen and the PI3K/HIF-1α/c-MYC/iNOS pathway in HCC are available. Our data show increased fibrinogen deposition in the peri-central zone and peri-central hepatocytes in NDEA-treated rats, accompanied by elevated levels of PI3K, HIF-1α, c-MYC, and iNOS, which were markedly reduced after SV high and low doses.

Of interest, the current finding demonstrated that a low dose of SV has similar or even better action against NDEA-induced HCC than a high dose of SV in most of the study results. SV extract contains a mixture of active and inactive components, potentially with high tissue exposure. Thus, SV extract’s lower potency components (beneficial or deleterious) may be in sufficient concentration at higher doses to begin to counteract the positive effects of its higher potency ligands. In addition, at high doses, the proportion of all components may increase so that one or more causes saturation of phase 1 metabolic enzymes (e.g., CYP450s), while at low doses, the proportions of active constituents are sufficient for notable activity without impairing biotransformation of SV components.

## 4. Materials and Methods

### 4.1. Chemicals, Kits, and Antibodies

Sigma-Aldrich Co. (St. Louis, MO, USA) was the source for acquiring NDEA. ELISA kits for measuring serum phosphoinositide 3-kinase (PI3K) and α-fetoprotein (AFP) were purchased from CUSABIO (Balitmore, MD, USA). Carcinoembryonic antigen (CEA) was purchased from Elabscience (Houston, TX, USA). VEGFR and EGFR proteins were purchased from Cloud-Clone Crop (Houston, TX, USA). Mouse monoclonal antibodies for fibrin and iNOS were ordered from Santa Cruz Biotechnology (Dallas, TX, USA). Invitrogen Thermo Fisher Scientific (San Francisco, CA, USA) provided the goat anti-mouse Cy3 and the anti-rabbit IgG horse radish peroxide (HRP) secondary antibodies. In addition, calorimetric assay kits for measurement of MDA production, GSH content, SOD, and catalase activities (Bio-diagnostic Co., Giza, Egypt) were utilized. Other solvents and compounds utilized in this investigation were of high analytical quality.

### 4.2. Plant Material

Aerial parts of the plant were collected from the Qassim region in Saudi Arabia during spring 2021, and its identity was confirmed to be *Suaeda vermiculata* Forssk. ex JF Gmel. by the botanist Ibrahim Aldakhil. The plant was stored under voucher #78 at the College of Pharmacy, Qassim University. Plant material was shade-dried, pulverized, and then extracted with 95% methanol for 24 h. The extract was filtrated and evaporated to dryness, adopting rotavapor, and stored at −20 °C while waiting for biological exterminations.

### 4.3. Animals

Adult male albino rats (180–220 g) were randomly selected for the current study. At the start of this trial, the animals were around 15 weeks old (provided by the animal house at Nahda University in Beni-Suef, Egypt). At the beginning of the experiment, rats were acclimatized for 1 week and kept at a humidity of 55, a temperature of 25 °C, and a 12 h light/dark cycle interval. Rats were kept on regular food pellets obtained from El-Nasr Co. (Abou-Zaabal, Egypt), with unlimited tap water access or 0.01% NDEA. The experimental methodology followed Nahda University criteria and was authorized by the university’s ethical review board in Beni-Suef, Egypt (ethical approval number NUB 025-022).

### 4.4. Experimental Study Protocol

Rats were classified randomly into 5 groups (12 animals each) and kept for 1 week for acclimatization. As the trial progressed, Group I received saline orally and served as a control group. Animals in Group 2 (SV control group) received SV p.o. (200 mg/kg b. wt.) in 1% of tween 80 and normal saline every day for 16 consecutive weeks. Rats in Group 3 (untreated group of animals) received freshly prepared NDEA solution (100 mg/L, Sigma-Aldrich Co., St. Louis, MO, USA) for the first 8 consecutive weeks of the experiment [46].

Groups 4 and 5 received NDEA + SV p.o. at 100 mg/kg b. wt. (NDEA + SV LD) and NDEA + SV p.o. at 200 mg/kg b. wt. (NDEA + SV HD), respectively. Both groups received NDEA and SV simultaneously for 16 weeks.

At the end of the experiment, after the animals had fasted overnight, blood was drawn from the orbital plexus under anesthesia with diethyl ether and then they were euthanized by decapitation. The collected blood was left for 30 min at 25 °C to coagulate before being centrifuged to prepare the serum. Liver samples were collected after using anesthesia to sacrifice the rats and then washed in saline. Pieces of liver were homogenized in PBS (pH 7.4) and centrifuged at 4000 rpm for 20 min under cooling conditions. The supernatants resulting from centrifugation were divided into aliquots and kept at −20 °C until biochemical assays. On the other hand, a suitable piece of liver from all groups was fixed in Davidson’s solution for immunofluorescence and immunohistochemistry examinations.

### 4.5. Assessment of Serum Tumor Biomarkers

The levels of AFP and CEA were measured in the animals’ serum using ELISA kits, which were processed according to the supplier’s instructions.

### 4.6. Assessment of the Products of Polyunsaturated Fatty Acids Peroxidation, MDA, and Antioxidant Biomarkers

ELISA kits from Bioassay Technology Laboratory (Jiaxing, China) were used to measure the levels of MDA in addition to the activities of CAT, SOD, and GSH enzymes in liver tissue homogenates following the manufacturer’s instructions.

### 4.7. Assessment of EGFR, PI3K, and VEGFR Levels

EGFR, VGFR, and PI3K serum levels were assessed using the ELISA technique and following the instructions of the manufacturers.

### 4.8. RNA Extraction for Real Time-PCR

TRI REAGENT Kits for RNA extraction from liver tissues were obtained from Thermo Fisher Inc. (New York, NY, USA), Cat. No. TR118. Using a Nanodrop-1000 spectrophotometer, the RNA content was measured at 260–280 nm (OD260) optical density. qRT-PCR was performed using samples with a 260/230 ratio greater than 1.5 and a 260/280 ratio greater than 1.8. The primers used in the study were HIF-1a forward—5′-GGC GCG AAC GAC AAG AAA AA-3′ reverse—5′-GTG GCA ACT GAT GAG CAA GC-3′; c-MYC–Forward 5′-ATC ACA GCC CTC ACT CAC-3-Reverse-5-ACA GAT TCC ACA AGG TGC-3 and GAPDH forward–5′ GTA TTG GGC GCC TGG TCA CC-3′ reverse—5′-CGC TCC TGG AAG ATG GTG ATG G-3′. In the current study, a real-time reverse transcription (RT)-PCR technique on a ViiA7 real-time PCR system (Applied Biosystems, Carlsbad, CA, USA) was employed with the aid of TOPreal™ qPCR 2× PreMIX (SYBR Green with low ROX; Enzynomics, Daejeon, Republic of Korea). Evaluation of qRT-PCR was conducted utilizing a master mix from SYBR green dye (RealMOD Green Real-time PCR 2× Master Mix Kit, iNtRon) purchased from Merck (Seoul, Republic of Korea) and the ABI StepOnePlus™ Real-Time PCR System purchased from Applied Biosystems (San Francisco, CA, USA). Levels of mRNA were computed and subsequently standardized to glyceraldehyde 3-phosphate dehydrogenase (GAPDH) using the 2^−ΔΔCt^ formula. The resulting values were then presented as a percentage relative to the reference gene.

### 4.9. Fluorescence Microscopic Examination

The method of immunofluorescence technique for fibrinogen expression was performed according to the modification of Abdel-Bakky 2022 [47]. After fixation in Davidson’s solution, liver tissues were embedded in paraffin wax and cut at 4 µm thickness. Tissues were antigen-retrieved by boiling in a buffered citrate solution (pH 6.0) for 20 min using the microwave after being deparaffinized and rehydrated. Slides were blocked using horse serum (10%) and BSA (1%) in PBS for 60 min at RT. Tissues were incubated overnight with a diluted blocking solution containing antibodies (anti-fibrinogen mouse monoclonal primary antibody, 1:150). Slides were washed in tween 20 (0.05%) in PBS and incubated with conjugated Cy3 anti-mouse secondary Ab (1:300 dilutions in blocking buffer) for 30 min. Slides were then incubated with DAPI (4,6-diamidino-2-phenylindole), and figures were captured and analyzed using a Leica DM 5500B epifluorescence microscope (Wetzlar, Germany). For measurement of the fluorescence intensity, at least 3 animal tissue samples were used from each group and 15 different field images from each section were recorded using Image-J/NIH software. Mean pixel intensity (from 0 to 255 scale values, next subtraction of the background) was used and calculated using GraphPad Prism 8.

### 4.10. Immunohistochemical Analysis

Paraffin tissue samples were super-frosted, then baked for 20 min at 60 °C to liquify paraffin, followed by two 15 min deparaffinizations in 100% xylol. A rehydration step was carried out utilizing graded ethanol series (5 min each): twice in 100%, once in 90%, once in 70%, 50%, once in 30%, and then in water. Slides were antigen retrieved by boiling in citrate buffer before being rinsed for 5 min in distilled water. Sections were kept for 10 min in methanol containing 3% H_2_O_2_ and were incubated with 10% normal goat serum in PBS for 15 min. Slides were then incubated with mouse monoclonal iNOS (diluted 1:200) antibody for 24 h at 4 °C, then for 20 min at RT with IgG horse radish peroxidase (HRP) anti-rabbit secondary antibody. Hematoxylin was employed as a counterstain, and peroxidase activity was processed by developing the color through the utilization of diaminobenzidine (DAB). The effectiveness of immunostaining was determined using a Leica DM 2500 light microscope (New York, NY, USA).

### 4.11. Statistical Investigation

To analyze data, GraphPad Prism 8 was used, and the data were displayed as mean ± SEM and one-way ANOVA followed by a Tukey-Kramer post hoc test. Where *p* < 0.05 was considered significant. The normality of the data was confirmed using the Shapiro-Wilk, D’Agostino, and Pearson Anderson–Darling tests.

## 5. Conclusions

In conclusion, our data demonstrated that increased liver levels of PI3K, HIF-1α, c-MYC, and iNOS were found in NDEA-induced HCC cells in rats. Furthermore, SV successfully having a significant protective effect against HCC induced by NDEA could be through inhibition of the PI3K/HIF-1α/c-MYC/iNOS pathway. Using SV extract may serve as an important option for patients suffering from HCC, especially those with fast growth and hypoxic conditions.

## Figures and Tables

**Figure 1 pharmaceuticals-16-01470-f001:**
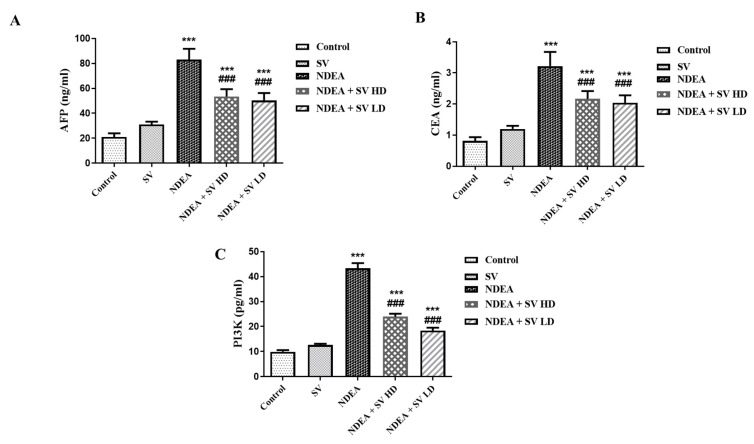
Effect of SV (100 or 200 mg/kg) on alpha-fetoprotein (AFP) (**A**), carcinoembryonic antigen (CEA) (**B**), and PI3K (**C**) serum levels in NDEA-induced rats’ hepatocellular carcinoma. Data represent mean + SEM using ANOVA and Tukey Kramer post-ANOVA. Where *** *p* < 0.001 is considered statistically significant compared to normal rats; ^###^
*p* < 0.001 is considered statistically significant compared to NDEA-treated rats. SV, *S. vermiculata* extract; NDEA, N-diethylnitrosamine; NDEA + SV HD, N-diethylnitrosamine plus *S. vermiculata* extract (200 mg/kg); NDEA + SV LD, N-diethylnitrosamine plus *S. vermiculata* extract (100 mg/kg).

**Figure 2 pharmaceuticals-16-01470-f002:**
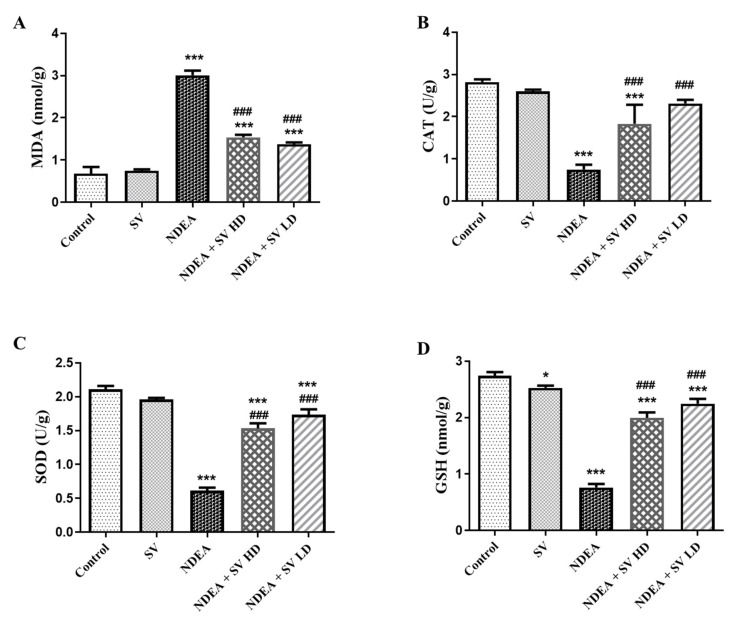
Effect of p.o. SV (100 or 200 mg/kg) on MDA (**A**); the anti-oxidant markers CAT (**B**), SOD (**C**), and GSH (**D**) levels in NDEA (200 mg/kg. p.o.)-induced HCC in rats’ liver tissue homogenates. Data represent mean + SEM using ANOVA and Tukey Kramer post-ANOVA. Where * *p* < 0.05 and *** *p* < 0.001 are considered statistically significant compared to normal rats; ^###^
*p* < 0.001 is considered statistically significant compared to NDEA-treated rats. SV, *S. vermiculata* extract; NDEA, N-diethylnitrosamine; NDEA + SV HD, N-diethylnitrosamine plus *S. vermiculata* extract (200 mg/kg); NDEA + SV LD, N-diethylnitrosamine plus *S. vermiculata* extract (100 mg/kg).

**Figure 3 pharmaceuticals-16-01470-f003:**
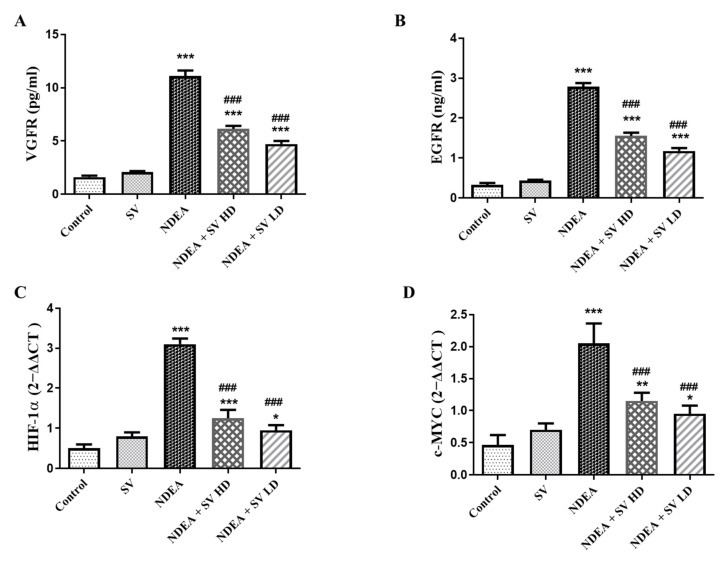
Effect of SV (p.o., high dose, 200 mg/kg, and low dose 100 mg/kg) on serum levels of EGFR (**A**), VEGFR (**B**), gene expression of HIF-1α (**C**), and c-MYC (**D**) in NDEA (200 mg/kg) caused rats HCC. Data represent mean + SEM using ANOVA and Tukey Kramer post-ANOVA as multiple comparisons between groups. Where * *p* < 0.05, ** *p* < 0.01, and *** *p* < 0.001 are considered statistically significant from normal rats; ^###^
*p* < 0.001 is considered statistically significant from NDEA-administered rats. SV, *S. vermiculata* extract; NDEA, N-diethylnitrosamine; NDEA + SV HD, N-diethylnitrosamine plus *S. vermiculata* extract (200 mg/kg); NDEA + SV LD, N-diethylnitrosamine plus *S. vermiculata* extract (100 mg/kg).

**Figure 4 pharmaceuticals-16-01470-f004:**
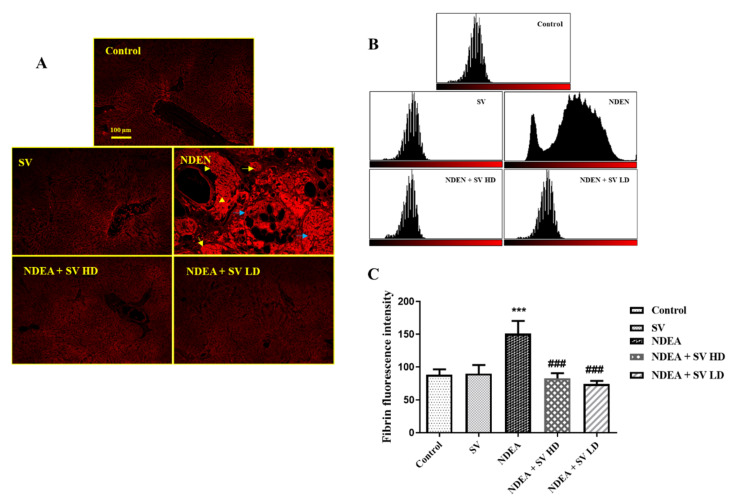
Effect of SV and/or NDEA on the liver tissue protein expression of fibrinogen. Rats treated with NDEA with SV (high or low doses) displayed noticeable reduction in fibrinogen expression to the basal level in the peri-central zone and peri-central hepatocytes compared to NDEA-only-treated rats (**A**); tissue expression histogram displaying the result of NDEA + SV HD and NDEA + SV HD on fibrinogen expression (**B**); and graphical quantification of fibrinogen expression resulted from fluorescence intensity, which was measured using Image-J/NIH software (https://imagej.nih.gov/ij/download.html) (**C**). *** *p* < 0.001 differs significantly from normal rats, ^###^
*p* < 0.001 differs significantly from NDEA-treated rats. Scale bar = 100 μm. Constitutive expression of fibrinogen in the livers of normal and SV-only-treated rats in peri-central hepatocytes and in the vascular epithelial cells. NDEA-treated rats showing accumulation of fibrinogen protein in the hepatic nodules (yellow arrows), large nodules of acinar HCC (blue arrows), and in the peri-central hepatocytes (arrowhead). SV, *S. vermiculata* extract; NDEA, N-diethylnitrosamine; NDEA + SV HD, N-diethylnitrosamine plus *S. vermiculata* extract (200 mg/kg); NDEA + SV LD, N-diethylnitrosamine plus *S. vermiculata* extract (100 mg/kg).

**Figure 5 pharmaceuticals-16-01470-f005:**
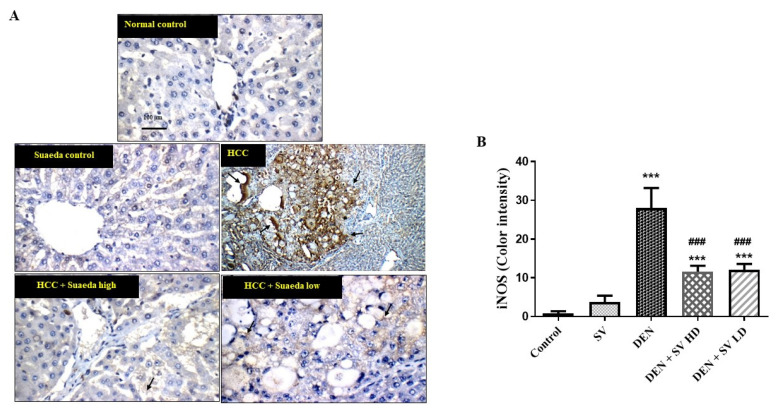
Effect of SV extract and/or NDEA on the liver tissue protein expression of iNOS figure (**A**) and color intensity (**B**). Tissues from control or SV control showing negative expression of iNOS. Rats treated with NDEA demonstrating increased expression of iNOS in the tumor cells in hepatic nodules (black arrows). Rats treated with NDEA + SV HD and NDEA + SV HD displaying noticeable reduction in iNOS expression in tumor cells of liver compared to NDEA-alone-administered rats. Where *** *p* < 0.001 are considered statistically significant compared to normal rats; ^###^
*p* < 0.001 is considered statistically significant compared to NDEA-treated rats. Scale bar = 100 μm. SV, *S. vermiculata* extract; NDEA, N-diethylnitrosamine; NDEA + SV HD, N-diethylnitrosamine plus *S. vermiculata* extract (200 mg/kg); NDEA + SV LD, N-diethylnitrosamine plus *S. vermiculata* extract (100 mg/kg).

## Data Availability

Data is contained within the article.

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
