# Peer review of "Involvement of PI3K/HIF-1α/c-MYC/iNOS Pathway in the Anticancer Effect of Suaeda vermiculata in Rats"

_pharmaceuticals, 2023, doi:10.3390/ph16101470_

Round 1

Reviewer 1 Report

In this in vivo work the authors tested the protective effect of Suaeda vermiculata (SV), at two different doses, on serum AFP, CEA, EGFR, PI3K and VEGFR levels, on hepatic MDA (pro-oxidant), CAT, SOD and GSH (antioxidants) levels in DENA-induced HCC rats. Gene expression of HIF-1α and c-MYC has been also evaluated.

The authors found that the treatment of the rats with DENA resulted in HCC induction and increased oxidative stress and reduced endogenous antioxidants. These effects decreased using high and low doses of SV.

The results obtained are interesting however some aspects of the work must be integrated and/or improved.

Minor revisions:

1)  Table 1 data should be plotted as in figure 1.

2) Figure 3. There is no match between what is shown in histogram “DENA+SVE LD” and the corresponding IF image “HCC + Suaeda low". The authors must replace the indicated IF image with a more representative one. Details of graphical quantification of fibrinogen expression must be reported in the relative M&M section. 

3) The analysis done for iNOS is purely qualitative. The authors have to quantify the expression with Image J software as they done for the fibrinogen expression.

4) Conclusion section has to be modified. The sentence “our data highlight a role of EGFR and VEGFR in activating PI3K, HIF-1α, c-MYC, iNOS thereby inducing HCC cell proliferation” is not supported by the data reported in this work and must be eliminated. With their experimental tests, the authors were able to verify “only” that SV treatment reduces these molecules levels induced by DENA.

5) The ways of indicating the treatments must be harmonized. The authors used SV or SEV or Suaeda or SVE random. I suggest to use SV in all charts, table, axis labels, figures, legends ecc.

6) Likewise, HCC and DENA have been used interchangeably. In these cases it is more correct to indicate the treatment rather than the effect therefore replace HCC with DENA in charts, table, axis labels, figures, legends ecc.

7) Uniform everything also as regards the indication of doses. HD or high? LD or low?

8) The caption of table 1 is not exhaustive. The doses HD (200mg/kg) and LD (100mg/kg) must be specified.

9) Fig 2. For C and D indicate the unit of measurement on the Y axis.

I am not particularly qualify to assess the quality of English in this paper but I have however found several errors.

Author Response

The Authors would like to thank and appreciate the efforts of reviewer 1 and his valuable comments on the manuscript. Attached below is point by point response to the reviewer’s comments.

Reviewer 2 Report

The authors report the chemoprotective effects produced by ingested Suaeda vermiculata Forssk. ex JF Gmel. (SV) in rats afflicted with hepatocellular carcinoma (HCC) induced by administration of the established N-nitrosoalkylamine carcinogen NDEA (i.e., DENA as designated by the authors). Both low dose and high dose SV had comparable effects, although curiously, the low dose SV appeared to be slightly superior in all studies. Are the authors able to rationalize this observation? Might it be associated with an oral bioavailability threshold of active components in SV or, alternatively, SV phase 1 metabolites actually being the active compounds (e.g., the high dose may limit CYP processing of soft prodrug SV compounds due to CYP saturation by less active SV components at higher concentrations)?

 Other key issues include:

·         All acronyms should be defined upon their first use in the manuscript. Some, such as MDA, are not defined until the end of the manuscript or even at all.

·         The manuscript is not written in idiomatic English, and particular paragraphs are difficult to comprehend (page 10 is challenging). Significant formatting and punctuation errors are also present throughout.

·         Most of the cited manuscripts in the Introduction are 10 years old or older and derived from regional authors (including excessive self-citations in low impact journals). The concepts described are general and popular. There are contemporary publications derived from authors worldwide associated with the topics. The references should be changed to include recent publications from those both in and outside of their region.

·         In the abstract, the authors state: “Herbal medicines have become an important alternative to chemotherapy drugs against liver cancer, this might be attributed to their fewer side effects and/or higher potency.” Herbal medicines do not have higher potency than single component drugs designed specifically as cancer treatments—at least not prior to development of API drug resistance—however, herbal medicines may have multiple active compounds that work cooperatively or synergistically to improve efficiency and reduce the possibility for early drug resistance.

·         On page 1, the authors state: “Liver cells including hepatocytes, are the most sensitive cells in the body to toxins…” This is not generally accepted as true. Lymphocytes and most brain and reproductive cells are far more sensitive to toxins than are hepatocytes.

·         Is DENA an accepted acronym for N-Nitrosodiethylamine? The common (only?) acronym is NDEA.

·         HD and LD, as shown in most table entries should be defined in each table heading or footnote.

·         MDA is not a “pro-oxidant” as defined throughout the manuscript (e.g., see bottom paragraph on p 5 for a couple instances and Figure 5 footnote, although other examples follow on later pages). MDA is a marker of oxidative stress. MDA does not promote oxidation—it is a metabolic byproduct of lipid oxidation.

·         Correct ‘rates’ to “rats” (pages 7, 8, and 9). The authors results are incomprehensible when the authors describe SV effects on rates rather than on the treated rats.

·         On page 10, the authors state: “Furthermore, DENA increased ox-idative stress through hepatic metabolism by cytochrome p450 enzyme (Anoopraj et al., 2014).” Although that may be possible, the authors have no experimental evidence for such a process. This statement should be revised to reflect this as a documented possibility only.

·         On page 10, the authors state: “Notably, SV significantly inhib-ited EGFR and VEGFR induced by DENA-treatment suggesting the important of both re-ceptors in the SV hepatoprotective effect.” This is untrue although possible. There is no evidence provided by the authors to indicate specifically what the SV components and/or their metabolites are actually interacting with—only that SV administration results in changes of concentrations of various disease related proteins and stress markers (e.g., maybe they are interacting with nuclear transcription factors or epigenetic enzymes instead of the noted cell receptors). The authors demonstrate effect but report nothing that indicates mechanism of action. Therefore, they should  not make such claims  without qualifying them as possibilities only.

·         Similarly, in the Conclusion, the authors claim: “In conclusion, our data highlight a role of EGFR and VEGFR in activating PI3K, HIF-1α, c-MYC, iNOS thereby inducing HCC cell proliferation.” Again, this statement cannot be made based on the experimental data provided. They show no evidence EGFR and VEGFR activate those proteins. Just because SV administration affects EGFR and VEGFR and the other protein concentrations does not confirm any such effects are connected in any way. Correlation does not equal causation. This statement must be removed or modified, as it is not supported by direct experimental evidence herein.

Overall, SV administration to NDEA-induced HCC rats offers interesting changes to key cellular protein and stress markers compared with controls. Despite the authors claims, no mechanism of action can be established through these experiments alone. Major corrections to claims, grammar, phrasing, formatting, and references are necessary prior to publication. However, if those are thoughtfully completed, I might be inclined to recommend for publication—but not before.

Many tens of errors in usage, grammar, punctuation, and formatting. The manuscript is difficult to comprehend in places, including most table and figure captions and footnotes.

Author Response

The Authors would like to thank and appreciate the efforts of reviewer 2 on his valuable comments about the manuscript. Attached below is point by point response to the reviewer’s comments.

Round 2

Reviewer 2 Report

Overall, the authors did an excellent job addressing changes to the language, references, and formatting to greatly improve upon the initial submission. The revised document both reads better and is more scientifically sound, although there are still minor grammatical errors throughout.

One significant issue remains in the new paragraph added to page 11 that must be addressed:

[p 11] “Of interest, the current finding demonstrated that low dose of SV has similar action or even better against NDEA-induced HCC than high dose in most of the study results. This could be explained by the severity of the side effects and adverse drug reactions are substantially dose-related. Also, SV extract may contains components with high specificity and high tissue exposure. For this a low dose, rather than high, is needed to attain greater efficacy/safety with high success rate. Furthermore, SV extract is composed of mixture of active and inactive constituents, at high doses the proportion of inactive component may increase to the limit that causes saturation of phase 1 metabolizing enzyme (CYP450) while on the other hand, at low dose this proportion of active constituent become in its required effective level.”

Since the authors do not provide an SV dose-response cytotoxicity study at varied SV concentrations (hence, their initial claim is not scientifically validated) and their second propose justification is unclear as written (High specificity for what? To what effect? How does that make low dose SV more effective than high dose SV?), the new paragraph is neither clear nor scientifically sound. Perhaps it could be reworded as follows:

‘Of interest, the current finding demonstrated that low dose SV has similar or even better action against NDEA-induced HCC than high dose SV in most of the study results. SV extract contains a mixture of active and inactive components, potentially with high tissue exposure. Thus, SV extract lower potency components (beneficial or deleterious) may be in sufficient concentration at higher doses to begin to counteract the positive effects of its higher potency ligands. In addition, at high doses, the proportion of all components may increase so that one or more causes saturation of phase 1 metabolic enzymes (e.g., CYP450s), while at low doses, the proportions of active constituents is sufficient for notable activity without impairing biotransformation of SV components.’

I recommend publication after the authors provide suitable changes to the indicated paragraph (but not before.)

Much improved over the original version, although editing for grammar would improve readability further.

Author Response

Since the authors do not provide an SV dose-response cytotoxicity study at varied SV concentrations (hence, their initial claim is not scientifically validated) and their second propose justification is unclear as written (High specificity for what? To what effect? How does that make low dose SV more effective than high dose SV?), the new paragraph is neither clear nor scientifically sound.

Reply:

Thanks for the valuable comment. The reworded phrase really clarifies the meaning.

This paragraph has been modified accordingly